# Study of Novel Furocoumarin Derivatives on Anti-Vitiligo Activity, Molecular Docking and Mechanism of Action

**DOI:** 10.3390/ijms23147959

**Published:** 2022-07-19

**Authors:** Chao Niu, Deng Zang, Haji Akber Aisa

**Affiliations:** 1State Key Laboratory Basis of Xinjiang Indigenous Medicinal Plants Resource Utilization, CAS Key Labortory of Chemistry of Plant Resources in Arid Zone, Xinjiang Technical Institute of Physics and Chemistry, Chinese Academy of Sciences, Urumqi 830011, China; niuchao@ms.xjb.ac.cn (C.N.); zangdeng@ms.xjb.ac.cn (D.Z.); 2University of Chinese Academy of Sciences, Beijing 100049, China

**Keywords:** vitiligo, furocoumarin, melanogenesis, SAR, molecular docking, p38 MAPK, Akt/GSK3β/β-catenin signaling pathways

## Abstract

Vitiligo is a common chronic dermatological abnormality that afflicts tens of millions of people. Furocoumarins isolated from Uygur traditional medicinal material *Psoralen corylifolia* L. have been proven to be highly effective for the treatment of vitiligo. Although many furocoumarin derivatives with anti-vitiligo activity have been synthesized, their targets with respect to the disease are still ambiguous. Fortunately, the JAKs were identified as potential targets for the disease and its inhibitors have been proved to be effective in the treatment of vitiligo in many clinical trials. Thus, sixty-five benzene sulfonate and benzoate derivatives of furocoumarins (**7a**–**7ad**, **8a**–**8ag**) with superior anti-vitiligo activity targeting JAKs were designed and synthesized based on preliminary research. The SAR was characterized after the anti-vitiligo-activity evaluation in B16 cells. Twenty-two derivatives showed more potent effects on melanin synthesis in B16 cells than the positive control (8-MOP). Among them, compounds **7y** and **8** not only could increase melanin content, but they also improved the catecholase activity of tyrosinase in a concentration-dependent manner. The docking studies indicated that they were able to interact with amino acid residues in JAK1 and JAK2 via hydrogen bonds. Furthermore, candidate **8** showed a moderate inhibition of CXCL−10, which plays an important role in JAK–STAT signaling. The RT-PCR and Western blotting analyses illustrated that compounds **7y** and **8** promoted melanogenesis by activating the p38 MAPK and Akt/GSK-3β/β-catenin pathways, as well as increasing the expressions of the MITF and tyrosinase-family genes. Finally, furocoumarin derivative **8** was recognized as a promising candidate for the fight against the disease and worthy of further research *in vivo*.

## 1. Introduction

Vitiligo is a chronic inflammatory cutaneous disorder that infects over 50 million people in the world [1]. Localized white patches and grey hair are the most common symptoms of the disease [2]. Although not fatal, it spoils the appearance and causes heavy psychological harm to patients. Furthermore, people suffering from vitiligo are vulnerable to other autoimmune diseases, such as rheumatoid arthritis, hyperthyreosis, diabetes and lupus erythematosus [3]. Many hypotheses about its mechanism have been proposed, including autoimmunity, genetics, environment, psychology, deficiency of trace elements, etc. [4]. Over the years, more and more clinical research has supported the immune theory [5,6,7]. However, the destruction of melanocytes and the obstruction of melanin synthesis directly lead to vitiligo [8,9].

Melanin is produced by melanocytes through a process called melanogenesis, catalyzed by tyrosinase-related protein 1 (TRP-1) and tyrosinase-related protein 2 (TRP-2) [10]. In the updated ‘‘Raper–Mason’’ pathway, it has been proven that both L-Dopa and the L-tyrosine bind to their exclusive active site of tyrosinase and oxidize to dopaquinone, which ultimately leads to eumelanin and pheomelanin via a series of reactions [11]. L-Dopa, derived from L-tyrosine catalyzed by THI (tyrosine hydroxylase isoform I) in melanosomes, in turn activates met-tyrosinase and reduces Cu(II) to Cu(I) on the enzyme active site [12]. TRP-1 and TRP-2 have roles in the biosynthesis of melanin downstream of TYR. TRP-2 catalyzes the dopachrome rearrangement to form 5, 6-dihydroxyindole-2-carboxylic acid (DHICA) [13], while TRP-1 oxidizes DHICA to produce carboxylate indolequinon [14]. The tyrosinase-family genes (TYR, TRP-1 and TRP-2) are tightly regulated by the microphthalmia-associated transcription factor (MITF), an important transcription factor regulating TYR gene expression [15]. It can modulate melanocyte functions, including melanocyte differentiation, pigmentation, proliferation and cell survival [16,17]. Several signaling pathways are present that clarify the specific mechanism controlling melanin biosynthesis via the tyrosinase family.

JAK–STAT signaling has been recently identified as a novel target for cancer and immune diseases [18,19]. Following further studies, JAK inhibitors have been found to show enormous potential in the treatment of immune dermatosis, such as psoriasis, vitiligo, alopecia areata (AA), atopic dermatitis (AD) and lupus erythematosus (LE) [20,21,22]. Among them, tofacitinib [23,24] and ruxolitinib [25,26,27] have both been shown to be effective against vitiligo; they have been proven to weaken the immunological reaction and relieve inflammation at lesions, which facilitates the repigmentation of skin.

Furanocoumarins (psoralens), first found in *Psoralea corylifolia* L., improve repigmentation along with sunlight in vitiligo treatment [28,29]. Although psoralens are the most commonly used drugs in clinical settings, the unknown targets of these compounds plagued the mechanism exploration and drug development of vitiligo.

In this work, two new series ester derivatives of furocoumarin (sixty-three in total) were designed and synthesized based on previous studies. The SAR was characterized after anti-vitiligo-activity evaluation in B16 cells. Docking studies with JAKs were also performed, accompanied by studies of the inhibition of chemokine secretion and the signaling pathways of the most promising compounds.

## 2. Results

### 2.1. Synthesis of Target Compounds ***7*** and ***7a**–**d***, ***8*** and ***8a**–**8ag***

The target ester derivatives were synthesized as depicted in Figure 1. Firstly, resorcine was converted to 4-methylumbelliferone (**2**) with the Pechmann reaction [30]. The etherification of **2** and 2-bromoacetophenone with potassium carbonate gave intermediate **3**. Moreover, it was treated with 4% KOH ethanolic solution to achieve a cyclization reaction. The oxidation of 5-methyl-3-phenyl-*7H*-furo[3,2-*g*]chromen-7-one using selenium dioxide in toluene gave 5-carbaldehyde **5**. The reduction of aldehyde to alcohol **6** was performed in the presence of NaBH_4_. The esterification with different benzenesulfonic and benzoic acid catalyzed by triethylamine finally afforded a variety of compounds (**7** and **7a**–**d**, **8** and **8a**–**8ag**).

### 2.2. Anti-Vitiligo-Activity Assay

The stimulation effects of compounds on melanin synthesis in B16 cells were screened at 10 and 50 μM. As shown in Table 1, the introduction of substituents on benzene was beneficial for melanin synthesis compared with the un-substituted one (**7h**). The electron-withdrawing groups (EWGs), such as -F, -Cl, -Br, -CF_3_ and -OCF_3_, seemed to be preferable. Derivatives **7a**, **7e**, **7i**, **7m**–**7n**, **7t**, **7u**, **7w** and **7y**, which were substituted by -F, -CF_3_ and -OCF_3_, with values from 149.5% to 191.2%, were more potent than the positive control (8-MOP (148.4%)). Among the halogenated derivatives, the ones with -F exhibited higher activity than those with -Cl and -Br (**7a** > **7g** > **7f**, **7m** > **7l** > **7o**, **7n** > **7j** > **7k**). However, the introduction of a second -F to benzene strengthened their abilities (**7y** > **7a, 7m**), while a decrease happened when an extra electron-donating group (EDG) was settled (**7n** > **7ac**). For these -F and -CF_3_ substituted derivatives, the *para*-position mostly contributed to the efficacy (**7a** > **7****m**–**7n**, **7i** > **7t**–**7u**). Moreover, it was interesting that the bulky groups availed melanin production (**7r** > **7q** > **7s**).

As shown in Table 2, thirteen benzoate derivatives (**8**, **8a**, **8i**, **8u**, **8y**–**8ag**) with values from 120.3% to 304.9% were superior to 8-MOP (113.4%). The EDGs (-CH_3_, -OCH_3_) and a strong EWG (NO_2_) were unfavorable for the melanin synthesis (**8h** > **8b**, **8d**, **8p**, **8v**, **8x**; **8h** > **8p**, **8w**). The introduction of bulky groups had few effects on melanogenesis (**8c**, **8r** and **8s**). For the halogenated derivatives, a similar result was observed, in that derivatives with -F were the most effective (**8a** > **8f**, **8g**; **8n** > **8j**, **8k**). The position of the substituents on benzene also played an important role in their efficacy; the shift of -F from the *ortho*-(**8m**) or *meta*-position (**8n**) to the *para*-position led to **8a**, which increased the content of pigment. Generally speaking, the more halogen atoms (-F, -Cl) were introduced to benzene, the higher the yield of melanin was. Besides -F-substituted derivatives, derivatives with -CF_3_ and -OCF_3_ could also enhance melanin synthesis (**8e**, **8i**, **8u**, **8y**–**8z**). More strikingly, the replacement of the heterocyclic ring with benzene resulted in a remarkable improvement in melanogenesis (**8ag** > **8h**).

Subsequently, the effects of **7y** and **8** on cell cytotoxicity, morphology, the catecholase activity of tyrosinase and melanin synthesis in PIG3V and B16 cells were further studied. Firstly, the CCK-8 assay was performed to determine the cytotoxicity of the two compounds in B16 cells. It was revealed, as can be seen in Figure 2C and Figure 3C, that compounds **7y** and **8** had weak toxicity at 0–10 µM and 0–50 µM compared with the control. In Figure 2B and Figure 3B, no significant abnormalities in the morphology and number of cells were observed. Moreover, the melanosomes in the cells continued to proliferate with the increase in the concentration. Compounds **7y** and **8** obviously promoted melanin production and the catecholase activity of tyrosinase in a dose-dependent manner in B16 and PIG3V melanocytes (Figure 2D–F and Figure 3D–F). However, after treatment with 8 at 50 µM, the melanin content (315.6%) was nearly 2.5-fold compared with 8-MOP (127.3%). Considering the weak effects of many active compounds on catecholase activity explored by our group, it was gratifying that compound 8 (148.4%) was better than 8-MOP for this key enzyme in melanin biosynthesis (127.3%).

### 2.3. Docking Studies

Molecular docking studies were performed to analyze the interaction between active derivatives (**7y** and **8**) and JAK kinases (JAK1 and JAK2). They interacted with JAKs by binding to the ATP sites (Figure 4, Figure 5, Figure 6 and Figure 7).

For **7y** and JAK2, the SO_2_ group interacted with Lys857 in the hinge region via a hydrogen bond, and another hydrogen bond was created by -C=O of Leu932 (Figure 4A). The unsubstituted benzene, furan and pyran rings made a great number of hydrophobic interactions with Ser939, Gly935, Leu855 and Leu983. When bound to JAK1, O^16^ of OSO_2_ and H^37^ of furan interacted directly with hinge Leu959 and P-loop Gly882 through hydrogen bonds (Figure 5A). The skeleton of furocoumarin and 5-benzene produced numerous hydrophobic contacts with Val889, Leu1010, Ala906 and Leu881. Hydrogen bonds between residues Gly887, Lys908 and F^26^ on benzene were found in JAK2. The halogen-mediated hydrogen bonds between F on benzene, and Val863 and Gly884 were also observed in both JAK1 and -2, although the former was very weak, since the distance was 3.3 Å (Figure 4A and Figure 5A).

In the docking mode of compound 8 with JAK2, O^16^ of the ester group could form hydrogen bonds with Gly856, while the other two hydrogen bonds were seen between Tyr931 and Leu932 in the hinge region, and O^13^ and H^29^ of the furan ring. The hydrophobic interactions with Leu855, Val863, Ala880 and Gly935 appeared to be able to stabilize the protein–small-molecule complex (Figure 6A). As shown in Figure 7A, -C=O of the ester group in 8 had hydrogen-bond interactions with Ser963 in the hinge region of JAK1. The furan ring (O^13^ and H^29^) was involved in two hydrogen bonds, with hinge residues Leu959 and Phe958. Additionally, CH_2_ on pyran pointed to a P-loop, constituting a hydrogen bond with Leu881. The skeleton of compound **8** was sandwiched in the binding pocket and engaged in a large number of hydrophobic interactions with Leu881, Nal889, Ala906 and Leu1010. In addition, compounds **7y** and **8** had docking poses more similar to those of control PF-06700841 with JAK1 (Figure 5B and Figure 7B) than to Ruxolitinib with JAK2 (Figure 4B and Figure 6B).

### 2.4. Inhibition Activity against JAK-Mediated Chemokine Production

In the present study, we performed an enzyme-linked immunosorbent assay (ELISA) to explore the potential involvement of derivatives **7y** and **8** in the regulation of IFN-γ-induced chemokine CXCL−10 levels in the supernatants of human-skin HaCaT keratinocytes. Following the treatment of HaCaT with 10 ng/mL recombinant IFN-γ in the presence or absence of **7y** and **8** (0.1, 1 or 10 μM), the levels of CXCL−10 were examined, as shown in Figure 8. The treatment of cells with a combination of both IFN-γ and Ruxolitinib (positive control) resulted in a notable reduction in CXCL−10 expression (almost 35-fold) compared with IFN-γ alone. Derivative **8** suppressed the CXCL−10 levels (*p* <  0.001) in IFN-γ-stimulated HaCaT cells at 1 and 10 μM, which was consistent with the molecular-modeling results (Figure 8A). In addition, when IFN-γ was not added, no obvious effects of the compounds on CXCL−10 secretion were observed, probably because JAK–STAT signaling was not activated by IFN-γ (Figure 8B).

### 2.5. Studies of Mechanisms of Action of ***7y*** in Melanin Synthesis 

#### 2.5.1. Effects of **7y** on the Expression of Melanogenesis-Related Genes

With real-time PCR (RT-PCR) and Western blotting analyses, compound **7y** was studied for its promotion of the expression levels of melanogenesis-related genes. As demonstrated in Figure 9A–C, the protein and mRNA expressions of MITF, TYR and TRP-1 increased in a concentration-dependent manner in B16 cells, except for TRP-2, after treatment with **7y**. These results indicate that compound **7y** may enhance melanogenesis by greatly upregulating the MITF and TYR genes but not TRP-2.

#### 2.5.2. Regulation of Compound **7y** of p38 MAPK Signaling Pathway

The MAPK signaling pathway is well known for its regulation of the generation of melanin. Thus, the influence of **7y** on the activation of the MAPK signaling pathway was investigated. Judging from Figure 10A,B, compound **7y** elevated the phosphorylation of P38 MAPK in a dose-dependent manner, but it was not beneficial for ERK and JNK phosphorylation.

#### 2.5.3. Effects of p38 Inhibitors on Compound **7y**–Induced Melanogenesis

To further identify the involvement of the P38 MAPK signaling pathway in compound **7y**–induced melanogenesis, p38 inhibitor SB203580 was utilized to assess the catecholase activity of tyrosinase and melanin contents. It was found that the melanogenic effects (melanin content and catecholase activity) induced by **7y** obviously decreased with SB203580 (a p38MAPK inhibitor) but were not affected by SP600125 (a JNK inhibitor) nor PD98059 (an ERK inhibitor) (Figure 11).

### 2.6. Studies of Mechanisms of Action of ***8*** in Melanin Synthesis 

#### 2.6.1. Effects of **8** on the Expression of Melanogenesis-Related Genes

Firstly, the effects of compound **8** on the expression of melanogenesis-related proteins were examined using Western blotting. As shown in Figure 12A,B, the protein expressions of MITF, TYR, TRP-1 and TRP-2 increased significantly in a concentration-dependent manner in B16 cells treated with compound **8,** especially at 50 µM, which indicated that it could enhance melanogenesis by upregulating the MITF and TYR proteins.

After that, we studied the effects of **8** on the relative quantification of the mRNA transcriptions of the *tyr*, *trp 1* and *trp 2* genes with the aid of RT-PCR. It can be seen from Figure 12C that treatment with **8** remarkably enhanced the tyrosinase-family-protein mRNA levels, which suggests that it may also improve the transcription levels of these three melanogenesis-related genes.

#### 2.6.2. Regulation of Compound **8** of the Akt/GSK-3β/β-Catenin Signaling Pathway

In order to explore the mechanisms of **8** in promoting melanin synthesis in B16 cells, the effects of **8** on related signaling pathways were examined *in vitro*. The Western blot analysis was preformed to detect the expression levels of Akt, GSK-3β, β-catenin, JNK, p38, ERK and CREB. According to Figure 13A,B, the expressions of phosphorylated Akt, GSK-3β and total β-catenin increased in a dose-dependent manner after treatment with **8** at concentrations from 1 to 50 µM. However, the compound had little influence on the phosphorylation of JNK, p38, ERK or CREB.

#### 2.6.3. Effects of GSK-3β Inhibitor on Compound **8**–Induced Melanogenesis

To understand the involvement of the Akt/GSK-3β/β-catenin pathway in **8**–induced melanogenesis, an Akt inhibitor was employed for the evaluation of the catecholase activity of tyrosinase and melanin content. As illustrated in Figure 14, it was obvious that these two important biological indexes both markedly decreased when co-treated with the Akt inhibitor and compound **8** compared with the latter alone. Moreover, this results firmly confirms the participation of Akt/GSK-3β/β-catenin signaling in compound-**8**-induced melanogenesis.

## 3. Discussion

Pathways such as Akt/GSK-3β/β-catenin, p38 MAPK, ERK and JNK MAPK signaling and cAMP-dependent pathways have been reported to be involved in melanogenesis via the regulation of MITF and TYR expressions. On the other hand, the MITF is regarded as the most important transcription factor for melanocyte development and the production of melanin, since it can regulate melanogenesis-related gene transcription, including tyrosinase, TRP-1 and TRP-2. It this research study, compound **7y** could markedly ameliorate the melanogenic effect and increase the expressions of melanogenic enzymes (TYR and TRP-1) and MITF in a concentration-dependent manner in cells. Our results demonstrate the **7y**–induced promotion of the phosphorylation of p38 with the exceptions of ERK and JNK, which may be blocked by p38 inhibitors. Taken together, compound **7y** induced melanogenesis by activating the p38 MAPK pathway and increasing the expressions of the MITF and tyrosinase-family genes (Figure 15).

The Western blot and RT-PCR analyses were applied to clarify that compound **8** enhanced melanogenesis by elevating the expression level of the MITF. Moreover, it was found that the expressions of phosphorylated Akt, GSK-3β and β-catenin considerably increased in cells after treatment with compound 8. Therefore, we assume that the effects of **8** on melanin synthesis may result from the activation of p-Akt, which promotes the phosphorylation of GSK-3β, thus increasing the accumulation of β-catenin in the cytoplasm; the accumulated β-catenin is translocated to the nucleus, where it directly binds with the MITF to stimulate its transcription and then stimulates the tyrosinase-family genes (Figure 16). In addition, an Akt inhibitor could dramatically reduce the content of melanin and the catecholase activity of tyrosinase, meanwhile few effects of **8** on p38 MAPK, ERK or JNK MAPK signaling or cAMP-dependent pathways were observed, which implies that compound **8** improved melanogenesis by activating the Akt/GSK-3β/β-catenin pathway and increasing the expression of the MITF, as well as of tyrosinase-family genes.

The Janus kinase (JAK) family, composed of JAK1, JAK2, JAK3 and TYK2, plays a crucial rule in the activation, differentiation and homeostasis of immune-associated cells [31]. Numerous JAK inhibitors have been developed due to its attractive effect on immunological regulation. In a recent study [32], IFN-γ–JAK/STAT signaling was proven to be closely related with the progression and maintenance of vitiligo. Melanocyte-reactive CD8^+^ T cells produce IFN-γ upon the encounter of melanocyte antigen; IFN-γ activates JAKs in keratinocytes (KCs) by binding to its receptors on KCs. After dimerization, JAKs are activated to phosphorylate their downstream STAT, and phosphorylated STAT moves into the nucleus and produces more CXCL−9 and CXCL−10, which leads to the additional recruitment of CD8^+^ T cells to melanocytes through the CXCR3 chemokine receptor. Treatment that inhibits this pathway, such as JAK inhibitors and monoclonal antibody agents for IFN-γ or CXCL−10, could effectively reverse the disease. Moreover, FDA-approved JAK inhibitor ruxolitinib has already been tested in a phase III trial for vitiligo (clinical trial NCT04530344). Thus, our group focused on the development of a small-molecule inhibitor of JAKs and discovered several hits from quinazoline and quinoline alkaloids with the help of virtual screening. Multi-target therapeutics that regulated multiple nodes of the disease network simultaneously showed a synergistic effect, which was unlikely to induce resistance and provided optimal clinical use. Fortunately, furocoumarin derivative **8** could inhibit CXCL−10 secretion in a concentration-dependent manner, which means that it may not only activate Akt/GSK-3β/β-catenin signaling but also block the JAK–STAT pathway, in spite of structural differences from the reported JAK inhibitors. Further structure optimization and target validation based on compound **8** are needed to explore more candidates for vitiligo.

## 4. Materials and Methods

### 4.1. Chemistry

#### 4.1.1. General Methods

All the chemicals were commercially available and were used directly without purification. A thin-layer chromatography (TLC) analysis was performed on glass plates coated with silica gel (Qingdao Haiyang Chemical Co.; G60F-254) and visualized under UV light (254 nm). The products were purified using silica gel (Qingdao Haiyang Chemical Co.; 200–300 meshes) using column chromatography. The NMR spectra were recorded on Varian (400 and 600 MHz) spectrometers, referenced in CDCl_3_ to tetramethylsilane (TMS). The melting points were measured on Buchi B-540. High-resolution mass spectra were acquired on an ABSciex QSTAR Elite quadrupole-time-of-flight mass spectrometer.

#### 4.1.2. Synthetic Procedures

##### Synthesis of 7-Hydroxy-4-methyl-2*H*-chromen-2-one (**2**)

To an ice-cold solution of resorcinol (2.0 g; 18.2 mol) in dioxane, conc. H_2_SO_4_ (0.5 mL) was added dropwise at 20 °C. After that, ethyl acetoacetate (2.8 g; 21.8 mmol) was added, and the mixture was heated to 60 °C for 4 h. Then, the mixture was poured into cold water, and the precipitate was filtered and dried under reduced pressure. The resulting mixture was recrystallized from methanol to obtain **2** as white needle crystals. Yield of 92%, m.p. 202–204 °C.

##### Synthesis of 4-Methyl-7-(2-oxo-2-phenylethoxy)-2*H*-chromen-2-one (**3**)

A mixture of **2** (0.88 g; 5.0 mmol) with 2-Bromoacetophenone (1.49 g; 7.5 mmol) and anhydrous K_2_CO_3_ (1.4 g; 10 mmol) in acetone (50 mL) was refluxed with stirring for 4 h. After cooling, the reaction mixture was filtered, and the filtrate was evaporated under reduced pressure. The obtained residue was purified using silica gel chromatography with a mixture of petroleumether and ethylacetate to obtain intermediate **3**. Yield of 90%, white solid, m.p. 162–164 °C. ^1^H NMR (400 MHz; CDCl_3_) *δ* 7.98 (dd, *J* = 8.2, 1.0 Hz, 2H), 7.64 (t, *J* = 7.4 Hz, 1H), 7.51 (m, 7.55–7.48, 3H), 6.94 (dd, *J* = 8.8, 2.6 Hz, 1H), 6.79 (d, *J* = 2.5 Hz, 1H), 6.13 (d, *J* = 1.1 Hz, 1H), 5.38 (s, 2H), 2.38 (d, *J* = 1.1 Hz, 3H).

##### Synthesis of 5-Methyl-3-phenyl-7*H*-furo[3,2-g]chromen-7-one (**4**)

To an ethanolic solution (100 mL) of intermediate **3** (2.94 g; 10 mmol), we added 4% ethanol potassium hydroxide solution (20 mL), and the mixture was refluxed for 4 h. After cooling, the solution was acidified with 1M hydrochloric acid and extracted with ethylacetate three times. The organic phase was dried overnight and evaporated under reduced pressure. The resulting residue was purified using silica gel chromatography with a mixture of petroleumether and ethylacetate to obtain compound **4**. Yield of 95%, white solid, m.p. 171–173 °C. ^1^H NMR (400 MHz; CDCl_3_) *δ* 7.98 (s, 1H), 7.83 (s, 1H), 7.64 (dd, *J* = 8.2, 1.1 Hz, 2H), 7.50–7.56 (m, 3H), 7.44 (td, *J* = 7.4, 1.1 Hz, 1H), 6.29 (s, 1H), 2.52 (s, 3H).

##### Synthesis of 7-Oxo-3-phenyl-7*H*-furo[3,2-g]chromene-5-carbaldehyde (**5**)

Powdered SeO_2_ (3.33 g; 30 mmol) was added to a solution of **4** (5.52 g; 20 mmol) in 20 mL of hot dry xylene, and the mixture was refluxed for 24 h with vigorous stirring under nitrogen conditions. The reaction mixture was filtered to remove black Se, and the deep orange filtrate was allowed to stand overnight. Almost-pure crystals of **5** could be separated from the solution. Yield of 63%, yellow solid, m.p. 148–150 °C. ^1^H NMR (400 MHz; CDCl_3_) *δ* 10.14 (s, 1H), 9.10 (s, 1H), 7.87 (s, 1H), 7.66 (d, *J* = 7.4 Hz, 2H), 7.59–7.49 (m, 3H), 7.44 (t, *J* = 7.2 Hz, 1H), 6.86 (s, 1H).

##### Synthesis of 5-(Hydroxymethyl)-3-phenyl-7*H*-furo[3,2-g]chromen-7-one (**6**)

Compound **5** (5.80 g; 20 mmmol) was dissolved in ethanol (130 mL), and sodium borohydride (380 mg; 10.0 mmol) was added; the solution was stirred for 2 h at room temperature. Thereafter, the suspension was carefully hydrolyzed with 1M HCl (20 mL), diluted with H_2_O and extracted three times with DCM. The organic phase was washed with brine, dried over Na_2_SO_4_ and evaporated under reduced pressure. The residue was purified using flash chromatography on silica gel to afford alcohol **6**. Yield of 76%, white solid, m.p. 99–100 °C. ^1^H NMR (400 MHz; CDCl_3_) *δ* 7.85 (s, 1H), 7.79 (s, 1H), 7.59 (d, *J* = 7.3 Hz, 2H), 7.55–7.47 (m, 2H), 7.46–7.38 (m, 2H), 6.62 (s, 1H), 4.97 (s, 2H).

##### Synthesis of (7-Oxo-3-phenyl-7*H*-furo[3,2-g]chromen-5-yl)methyl Methanesulfonate (**7**)

To an ice-cold mixture of alcohol **6** (2.92 g; 10 mmol) and triethylamine (4.04 g; 40 mmol) in ethylacetate, methylsulfonylchloride (2.30 g; 20 mmol) was added dropwise at 0 °C. The reaction mixture was allowed to warm to room temperature, stirred for 2 h and diluted with water (20 mL). The organic layer was dried over MgSO_4_ and filtered. The residue was purified using flash chromatography on silica gel to obtain sulfonate **7**. Yield of 85%, m.p. 180–18 °C. ^1^H NMR (400 MHz; CDCl_3_) *δ* 7.91 (s, 1H), 7.87 (s, 1H), 7.64–7.60 (m, 2H), 7.59–7.51 (m, 3H), 7.47–7.41 (t, *J* = 7.3 Hz, 1H), 6.57 (s, 1H), 5.47 (d, *J* = 1.2 Hz, 2H), 3.16 (s, 3H). ^13^C NMR (101 MHz; CDCl_3_) *δ* 160.21, 152.00, 157.41, 147.21, 143.44, 130.68, 129.51, 128.43, 127.72, 124.61, 122.49, 114.84, 113.32, 113.22, 101.02, 65.30, 38.47. HRMS (ESI) calcd for C_19_H_15_O_6_S^+^ [M+H]^+^ 371.0584, found 371.0578.

##### Synthesis of **7a**–**7ad**

To an ice-cold mixture of alcohol **6** (2.92 g; 10 mmol) and triethylamine (4.04 g; 40 mmol) in ethylacetate, diverse benzenesulfonylchloride (20 mmol) was added dropwise at 0 °C. The reaction mixture was allowed to warm to room temperature, stirred until the reaction was complete and then diluted with water (20 mL). The organic layer was dried over MgSO_4_ and filtered. The residue was concentrated and purified using flash chromatography on silica gel to obtain benzene sulfonate **7a**–**7ad**. The NMR and spectrum can be found in Appendix A.

##### Synthesis of (7-Oxo-3-phenyl-7*H*-furo[3,2-g]chromen-5-yl)methyl Acetate (**8**)

To an ice-cold mixture of alcohol 6 (2.92 g; 10 mmol) and triethylamine (4.04 g; 40 mmol) in ethylacetate, acetyl chloride (1.58 g; 20 mmol) was added dropwise at 0 °C. The reaction mixture was allowed to warm to room temperature, stirred for 2 h and diluted with water (20 mL). The organic layer was dried over MgSO_4_, filtered and concentrated. Purification by ether/ethylacetate was performed to obtain acetate **8**. Yield of 87%, m.p. 167–169 °C. ^1^H NMR (400 MHz; CDCl_3_) *δ* 7.86 (d, *J* = 7.0 Hz, 2H), 7.63–7.58 (m, 2H), 7.57–7.51 (m, 3H), 7.45 (t, *J* = 7.3 Hz, 1H), 6.51 (s, 1H), 5.39 (s, 2H), 2.24 (s, 3H). ^13^C NMR (101 MHz; CDCl_3_) *δ* 170.28, 160.75, 157.25, 151.94, 149.49, 143.30, 130.86, 129.48, 128.35, 127.70, 124.37, 122.44, 114.67, 113.85, 111.52, 100.85, 61.39, 20.90. HRMS (ESI) calcd for C_20_H_15_O_5_^+^ [M+H]^+^ 335.0914, found 335.0922.

##### Synthesis of **8a**–**8ag**

To an ice-cold mixture of alcohol **6** (2.92 g; 10 mmol) and triethylamine (4.04 g; 40 mmol) in ethylacetate, diverse benzoylchloride (20 mmol) was added dropwise at 0 °C. The reaction mixture was allowed to warm to room temperature, stirred until the reaction was complete and then diluted with water (20 mL). The organic layer was dried over MgSO_4_ and filtered. The residue was concentrated and purified using flash chromatography on silica gel to obtain benzene sulfonate **8a**–**8ag**. The NMR and spectrum can be found in Appendix A.

### 4.2. Biology

#### 4.2.1. Cell Cultures

The murine B16 melanoma cell line was purchased from Chinese Academy of Sciences (Beijing, China). The cells were maintained in Dulbecco’s modified Eagle medium (DMEM; Gibco Life Technologies, France, Paris) supplemented with 10% (*v*/*v*) FBS, penicillin G (100 U/mL) and streptomycin (100 mg/mL) (Gibco-BRL, Grand Island, NY, USA) in 5% CO_2_ at 37 °C.

Human keratinocyte cell line HaCaT was purchased from China Center for Type Culture Collection (HaCaT; Cat# GDC106); the cells were cultured in DMEM (Gibco Life Technologies, Waltham, MA, USA) supplemented with 10% FBS (Thermo Fisher Scientific), penicillin G (100 U/mL) and streptomycin (100 µg/mL) (Gibco-BRL, Grand Island, NY, USA) and incubated at 37 °C in a humidified atmosphere containing 5% CO_2_.

#### 4.2.2. Cell-Viability Measurement

Cell viability was assayed by adding CCK-8 solution. Generally speaking, B16 cells were seeded in 96-well plates at a density of 8 × 10^3^ cells per well and were allowed to adhere for 24 h. The medium was replaced with medium containing samples diluted to the appropriate concentrations. The control cells were treated with DMSO at the final concentration of 0.1%. After 24 h, the culture medium of the cells was discarded; a volume of 10 μL of CCK-8 solution was added into each well, and cells were incubated at 37 °C for another 2 h. The absorbance was measured at 450 nm using Spectra Max M5 (Molecular Devices, San Diego, CA, USA). All assays were performed in triplicate. Absorbance of cells without treatment was regarded as 100% of cell survival. Cell viability was calculated using the following formula: cell viability (%) = (A_sample_/A_control_) × 100%.

#### 4.2.3. Melanin Measurement

B16 cells were seeded at a density of 2 × 10^5^ cells/well in a 6-well plate. After overnight incubation, the test samples were added to individual wells, and cells were incubated for 48 h and washed twice with ice-cold PBS. After cells lysed, the harvested cells were centrifuged, and the pellet was dissolved by adding 1 M NaOH, followed by incubation at 80 °C for 1 h. Each lysate (150 μL) was put in a 96-well microplate and measured spectro-photometrically at 405 nm with a multi-plate reader. The protein concentration of each sample was determined with *BCA* Protein Assay Kit (Biomed, Beijing, China). Intracellular melanin amounts were expressed as abs/μg protein and were shown as percentage values. The percentage values of the sample-treated cells were calculated with respect to the untreated cells.

#### 4.2.4. Assay of Catecholase Activity of Tyrosinase 

The assay of catecholase activity was carried out as previously described, with a slight modification. B16 cells were seeded in a 6-well plate at a density of 2 × 10^5^ cells per well and were allowed to attach for 24 h. Test samples were then added to individual wells. After 24 h of incubation, cells were washed twice with ice-cold PBS and lysed with 1% Triton X-100 solution containing 1% sodium deoxycholate for 30 min at −80 °C; then, each lysate was centrifuged at 12,000× *g* RPM for 15 min to obtain the supernatant. After protein quantification and adjustment, 90 μL of the supernatant was incubated in duplicate with 10 μL of freshly prepared substrate solution (10 mM L-DOPA) in a well of a 96-well plate. Then, the cells were incubated at 37 °C in the dark for 60 min. The absorbance was measured at 490 nm, and the values of the sample-treated cells were presented as percentages against the untreated cells.

#### 4.2.5. Measurement of CXCL−10 Release by HaCaT Keratinocytes

HaCaT cells were plated at a density of 4 × 10^5^ cells per well in a 6-well plate and treated with the indicated concentrations of the compounds with Human IFN-γ (PeproTech, Cranbury, NJ, USA) at 10 ng/mL for 24 h and 3 µM Ruxolitinib as the positive control. To remove cell debris, the cell-culture supernatant was centrifuged at 1000× *g* RPM for 10 min. The release levels of CXCL−10 were detected with a Human CXCL−10/IP-10 enzyme-linked immunosorbent assay (ELISA) kit, which was purchased from Absin (Shanghai, China), based on the manufacturer’s instructions. The absorbance was measured at 450 nm using a Spectra Max M5 microplate reader (Molecular Devices company, San Diego, CA, USA).

#### 4.2.6. Western Blot Analysis

B16 cells were treated with different concentrations of the tested compounds in a 6-well plate for 48 h. Cells were then lysed in cold RIPA (radio immunoprecipitation assay) lysis buffer (pH 7.4) containing protease and protease inhibitor cocktail (1 M 4-nitrophenyl phosphate disodium salt hexahydrate (PNPP), 1 M sodium fluoride (NaF), 10 mM phenylmethanesulfonylfluoride (PMSF), 100 mM benzamidine, 100 mM DL-Dithiothreitol (DTT), 200 mM sodium orthovanadate (OV)) for 30 min on ice. The lysates were centrifuged at 12,000× *g* rpm for 20 min at 4 °C before the supernatant was collected. The samples’ protein concentrations were measured with BCA Protein Assay Kit (Biomed, Beijing, China) and were separated by 10% SDS polyacrylamide gels and transferred onto polyvinylidene fluoride (PVDF) membranes (Merck Millipore Ltd., Billerica, MA, USA). Membranes were incubated with the primary antibodies at 4 °C overnight and then with horseradish-peroxidase-conjugated secondary antibodies for 1 h at room temperature. The targeted proteins were detected using ECL Western blotting detection reagents (GE Healthcare, Beijing, China) and were visualized using a ChemiDoc MP Imaging system (Bio-Rad Laboratories, Inc., Berkeley, CA, USA). All Western blot assays were performed in triplicate.

#### 4.2.7. Primer Sequence in Quantitative Real-Time PCR

Total cellular RNA was prepared from B16 cells treated with the tested compounds and isolated with TRIzol reagent in accordance with the manufacturer’s instructions. Quantitative PCR was performed to determine the expressions of target genes. The primers were as follows: forward 5′-GTCGTCACCCTGAAAATCCTAACT-3′ and reverse 5′-CATCGCATAAAACCTGATGGC-3′ for Tyr (111 bp); forward 5′-ACCCATTTGTCTCCCAATGA-3′ and reverse 5′-GTCCAATAGGTGCGTTTTCC-3′ for TRP-1 (130 bp); forward 5′-TACCATCTGTTGTGGCTGGA-3′ and reverse 5′-TGGGTCATCTTGCTG-3′ for TRP-2 (147 bp); forward 5′-AGTACAGGAGCTGGAGATG-3′ and reverse 5′-GTGAGATCCAGAGTTGTCGT-3′ for MITF (181 bp). β-Actin was used as an internal control in all cases, and its primer sequence was as follows: forward 5′-TCAAGA TCATTGCTCCTCCTG-3′ and reverse 5′-CTGCTTGCT GATCCA-CATCTG-3′ (59 bp). All determinations were performed three times. The reaction parameters were 95 °C for 10 min, followed by 40 cycles of 15 s at 95 °C for melting and 1 min at 60 °C for annealing. Real-time PCR was performed using an Applied Biosystems 7300 PCR machine (Applied Bioscience, Foster City, CA, USA). The results were normalized to the controls.

#### 4.2.8. Statistical Analysis

All results were expressed as mean ± SD, and statistical analyses were performed with one-way ANOVAs, followed by Tukey’s multiple comparisons tests. Statistical analyses were performed using GraphPad Prism 9 (La Jolla, CA, USA). *p*-values < 0.05 were considered to be statistically significant.

#### 4.2.9. Molecular Docking

Human JAK1 (PDB: 6DBN) [33] and JAK2 (PDB: 6VGL) [34] were downloaded from the Protein Data Bank (http://www.rcsb.org accessed on 9 November 2021). They were prepared by removing water molecules and adding hydrogen atoms using protein preparation wizard in Discovery Studio 2016. After energy minimization, the docking studies of compounds **7y** and **8** with JAK1 and JAK2 were performed via CDOCKER. Before docking, a sphere (with a 10 Å radius) around the template molecule was established as the binding site. For the simulated annealing, all the parameters were set as default. After the docking procedure, ten top-ranked ligand–receptor conformations were obtained, and the binding patterns of the docked molecules were visualized and analyzed according to the receptor–ligand interactions in Discovery Studio.

## 5. Conclusions

Furocoumarins have been used to treat vitiligo for a long time. Moreover, in this work, sixty-five novel ester derivatives of furocoumarin were designed, synthesized and evaluated for their anti-vitiligo activity. Further pharmacological studies let us evaluate two candidates, **7y** and **8** that were more potent than the positive control (8-MOP) on both melanin synthesis and the catecholase activity of tyrosinase. The possible binding mode of the compounds with JAKs were identified with molecular docking, together with the inhibitory effect on IFN-γ-induced CXCL−10 production observed in the HaCaT-cell experiment.

Western blotting and RT-PCR proved that **7y** could promote the phosphorylation of p38, but not ERK and JNK, which demonstrated that the activation of p38 MAPK may be its underlying mechanism. On the other hand, compound **8** could activate p-Akt, which increased the phosphorylation level of GSK-3β, thus inhibiting the degradation of β-catenin. The accumulated β-catenin was translocated to the nucleus and combined with the MITF to strengthen its transcription; then, it stimulated the tyrosinase-family genes. Further studies of **8** in animal models of vitiligo are required to evaluate its safety and efficacy.

## Figures and Tables

**Figure 1 ijms-23-07959-f001:**
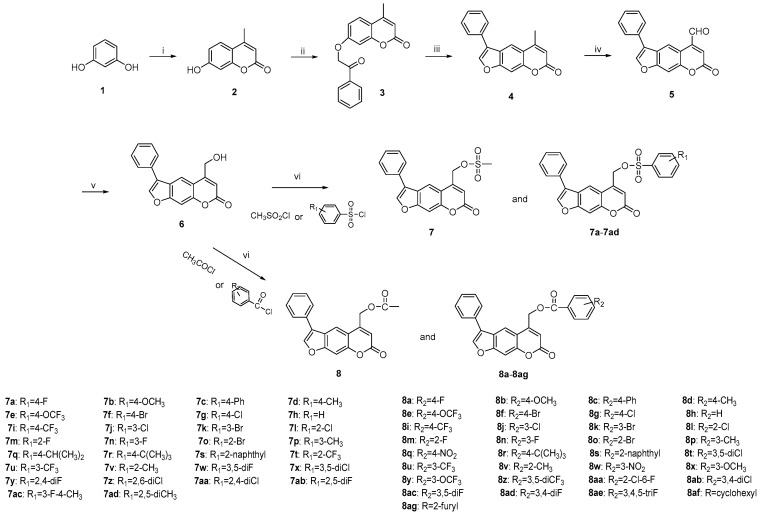
Synthesis of benzene sulfonate and benzoate derivatives of furanocoumarin. Reagents and conditions: (**i**) Ethyl acetoacetate, H_2_SO_4_; 60 °C. (**ii**) 2-Bromoacetophenone, K_2_CO_3_, acetone; reflux. (**iii**) 4% KOH ethanolic solution; reflux. (**iv**) SeO_2_, xylene; reflux. (**v**) NaBH_4_, methanol; rt. (**vi**) Et_3_N, DCM; rt.

**Figure 2 ijms-23-07959-f002:**
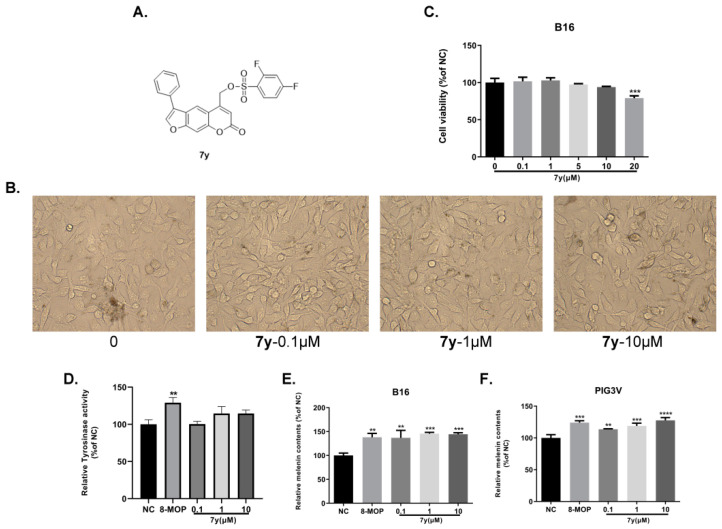
Dose-dependent effects of derivative **7y** on cellular melanin synthesis in B16 and PIG3V cells. (**A**) Chemical structure of **7y**. (**B**) Dose-dependent effect of **7y** on cell morphology in B16 cells. (**C**) Dose-dependent effect of **7y** on cell viability in B16 cells (*** *p* < 0.001; *n* = 3). (**D**) Effect of **7y** on the catecholase activity of tyrosinase in B16 cells. (**E**,**F**) Effect of **7y** on melanin content in B16 and PIG3V cells (** *p* < 0.01, *** *p* < 0.001, **** *p* < 0.0001 compared with untreated control cells; *n* = 3).

**Figure 3 ijms-23-07959-f003:**
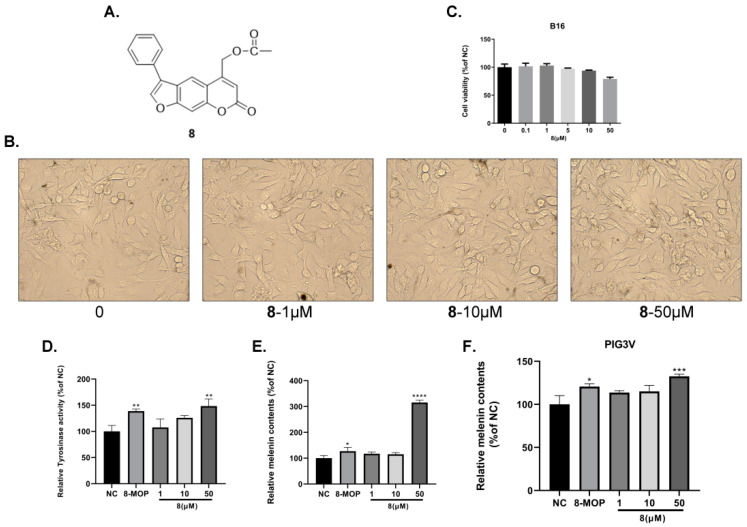
Dose-dependent effects of derivative **8** on cellular melanin synthesis in B16 and PIG3V cells. (**A**) Chemical structure of **8**. (**B**) Dose-dependent effect of **8** on cell morphology in B16 cells. (**C**) Dose-dependent effect of **8** on cell viability in B16 cells. (**D**) Effect of **8** on the catecholase activity of tyrosinase. (**E**,**F**) Effect of **8** on melanin content in B16 and PIG3V cells (* *p* < 0.05, ** *p* < 0.01, *** *p* < 0.001, **** *p <* 0.0001 compared with untreated control cells; *n* = 3).

**Figure 4 ijms-23-07959-f004:**
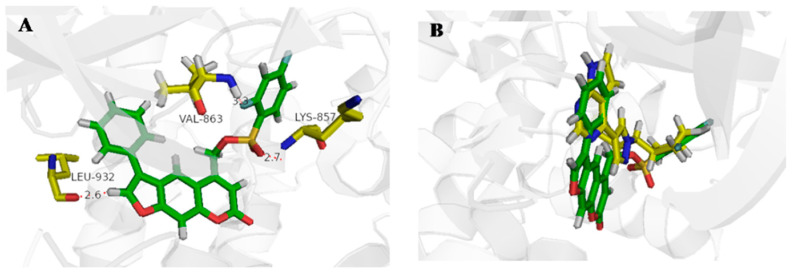
Docking result of **7y** with human JAK2. (**A**) Predicted binding mode of **7y** to JAK2. The red dots represented hydrogen bond (**B**) Overlap of **7y** (green) and Ruxolitinib (yellow).

**Figure 5 ijms-23-07959-f005:**
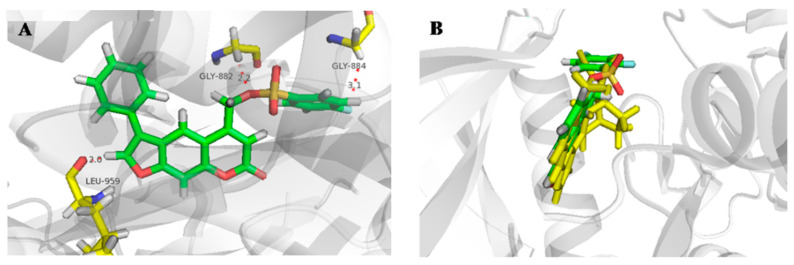
Docking result of **7y** with human JAK1. (**A**) Predicted binding mode of **7y** to JAK1. The red dots represented hydrogen bond (**B**) Overlap of **7y** (green) and PF-06700841 (yellow).

**Figure 6 ijms-23-07959-f006:**
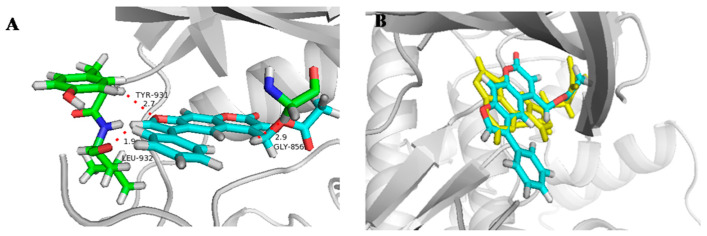
Docking result of **8** with human JAK2. (**A**) Predicted binding mode of **8** to JAK2. The red dots represented hydrogen bond (**B**) Overlap of **8** (light blue) and Ruxolitinib (yellow).

**Figure 7 ijms-23-07959-f007:**
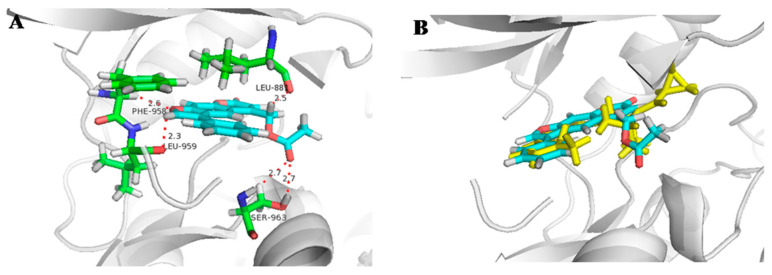
Docking result of **8** with human JAK1. (**A**) Predicted binding mode of **8** to JAK1. The red dots represented hydrogen bond (**B**) Overlap of **8** (light blue) and PF-06700841 (yellow).

**Figure 8 ijms-23-07959-f008:**
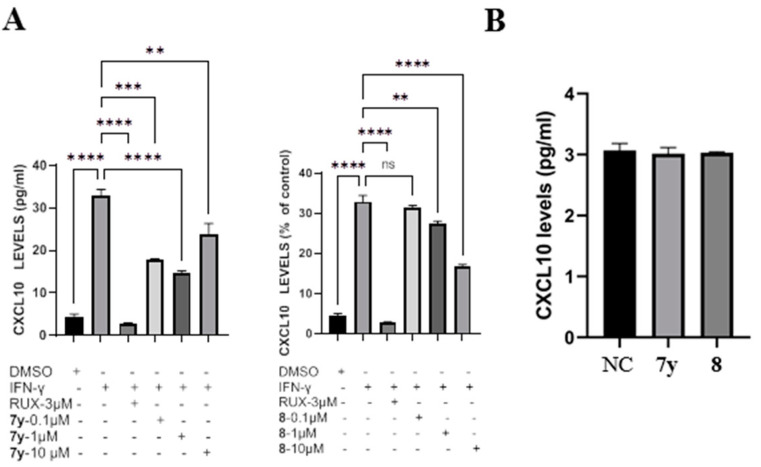
Effects of derivatives **7y** and **8** on CXCL−10 secretion in human-skin HaCaT keratinocytes. (**A**) Cells were pretreated with derivatives **7y** and **8** for 2 h before IFN-γ stimulation and were harvested 24 h later. (**B**) Cells were only treated with derivatives **7y** and **8**. The secretion levels of CXCL−10 in culture medium were determined with the ELISA kit. Each bar represents the mean ± SD (*n* = 3) (**** *p* < 0.0001, *** *p* < 0.001, ** *p* < 0.01 compared with IFN-γ−stimulated group; *n* = 3. ns means not significant).

**Figure 9 ijms-23-07959-f009:**
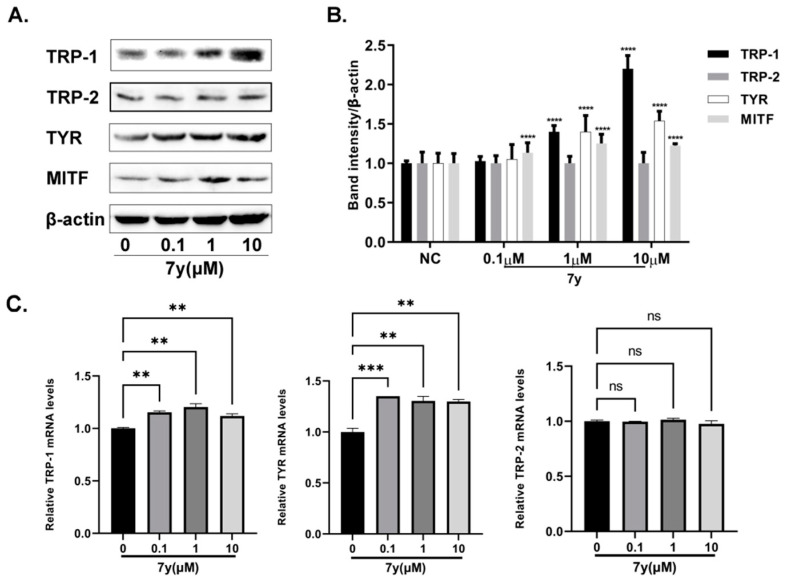
The effects of **7y** on TYR-protein content in B16 cells. (**A**) Effect of **7y** on tyrosinase-family-protein expression levels. (**B**) Protein-band expression levels (**** *p <* 0.0001 compared with untreated control cells; *n* = 3). (**C**) Effect of **7y** on tyrosinase-family-protein mRNA levels (** *p <* 0.01, *** *p <* 0.001 compared with untreated control cells; *n* = 3; ns means not significant).

**Figure 10 ijms-23-07959-f010:**
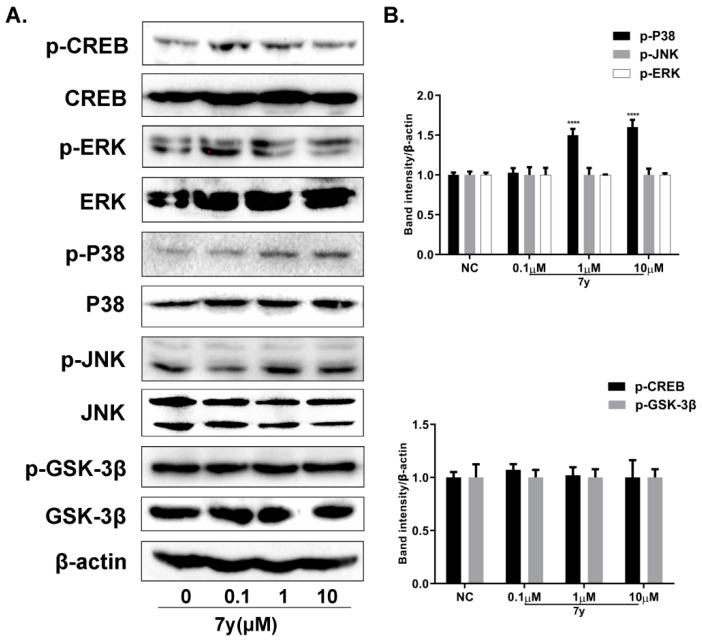
The effects of derivative **7y** on MAPK signaling pathway analyzed using Western blot assays. (**A**) Effect of **7y** on MAPK-signaling-protein expression levels. (**B**) Protein-band expression levels (**** *p* < 0.0001 compared with control group; *n* = 3).

**Figure 11 ijms-23-07959-f011:**
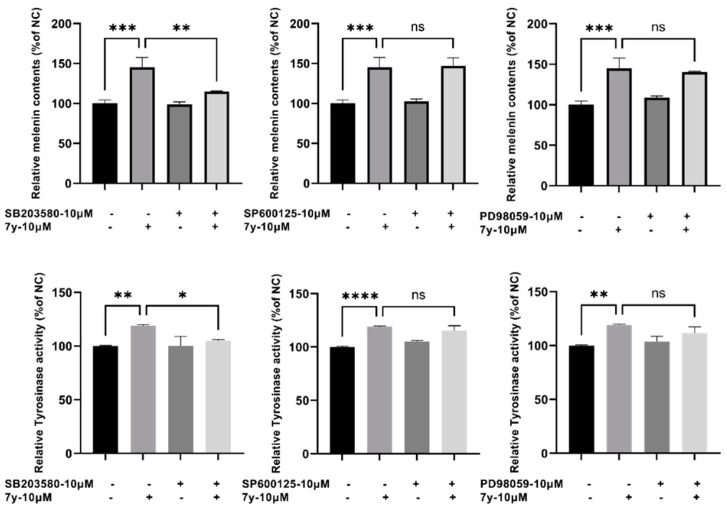
Influence of MAPK inhibitors on **7****y**–induced promotion of melanin content and catecholase activity of tyrosinase (p38 inhibitor; JNK inhibitor; ERK inhibitor; (* *p* < 0.05, ** *p <* 0.01, *** *p <* 0.001, **** *p* < 0.001 compared with control; *n* = 3; ns means not significant).

**Figure 12 ijms-23-07959-f012:**
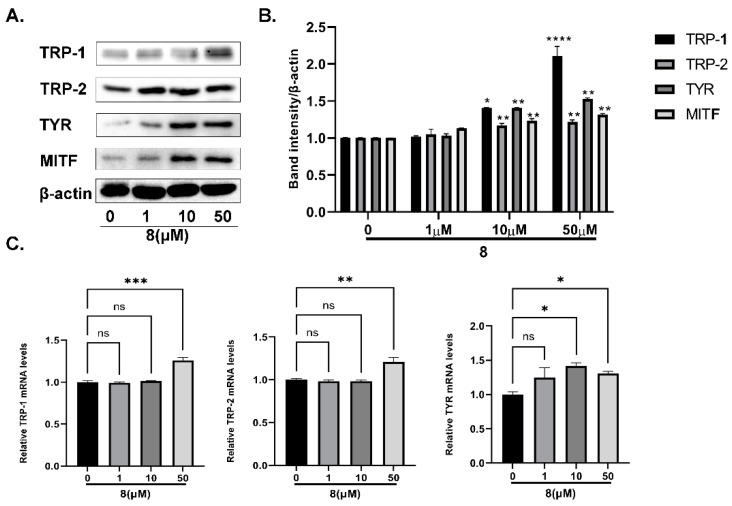
The effects of **8** on TYR-protein content in B16 cells. (**A**) Effect of **8** on tyrosinase-family-protein expression levels. (**B**) Protein-band expression levels (* *p* < 0.05, ** *p* < 0.01, **** *p <* 0.0001 compared with untreated control cells; *n* = 3). (**C**) Effect of **8** on tyrosinase-family-protein mRNA levels (* *p* < 0.05, ** *p <* 0.01,*** *p <* 0.001 compared with untreated control cells; *n* = 3. ns means not significant).

**Figure 13 ijms-23-07959-f013:**
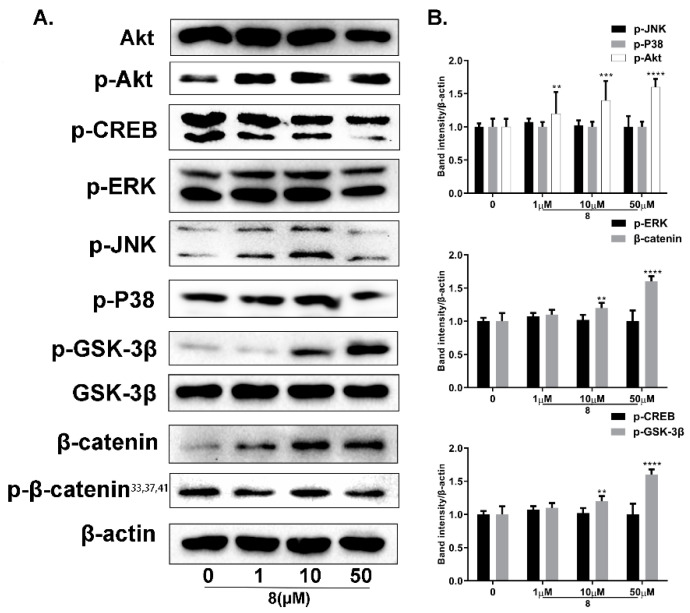
The effects of derivative **8** on GSK-3β/β-catenin signaling pathway analyzed using Western blot assays. (**A**) Effect of **8** on GSK-3β/β-catenin-signaling protein expression levels. (**B**) Protein-band expression levels (** *p <* 0.01,*** *p <* 0.001, **** *p* < 0.0001 compared with control group; *n* = 3).

**Figure 14 ijms-23-07959-f014:**
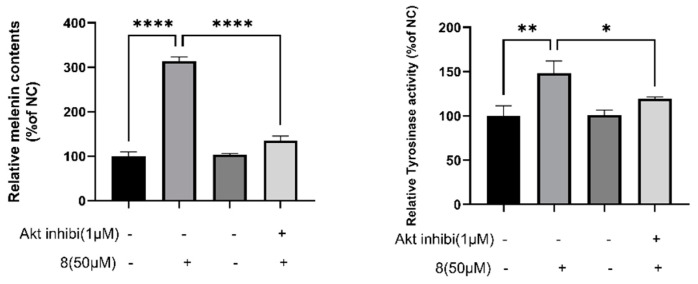
Influence of Akt inhibitor on derivative **8**–induced promotion of melanin content and catecholase activity of tyrosinase in B16 (* *p* < 0.05; ** *p* < 0.01; **** *p <* 0.0001, compared with **8** stimulation; *n* = 3).

**Figure 15 ijms-23-07959-f015:**
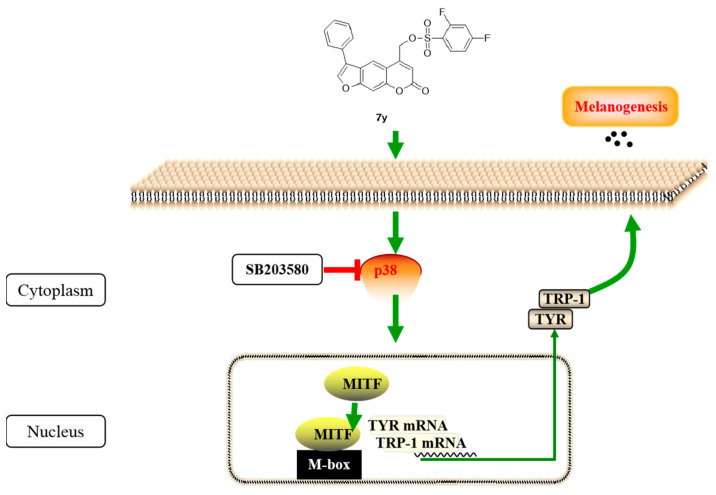
Mechanism of derivative **7y** in promoting melanogenesis *in vitro*.

**Figure 16 ijms-23-07959-f016:**
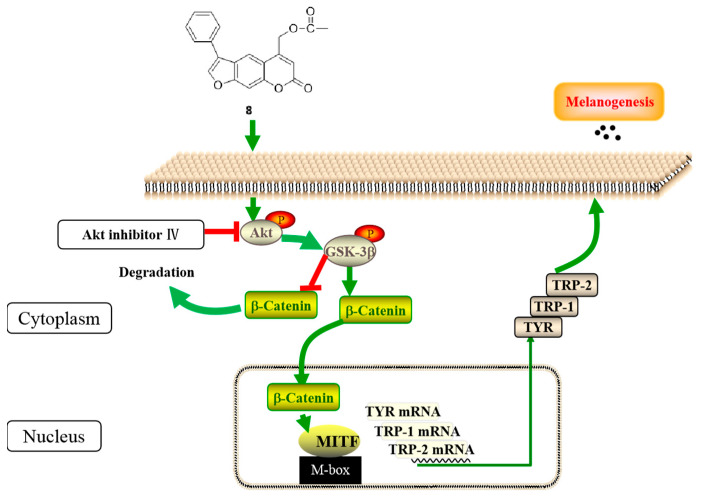
The mechanism of derivative **8** in promoting melanogenesis *in vitro*.

**Table 1 ijms-23-07959-t001:** The effects of derivatives **7** and **7a****–****7ad** on melanogenesis in B16 cells (10 μM).

Derivative	Substituent (R1)	Relative Melanin Content (%)
NC	-	100 ± 4
8-MOP	-	148 ± 4
**7**	-	126 ± 9
**7a**	4-F	164 ± 5
**7b**	4-OCH_3_	139 ± 5
**7c**	4-Ph	139 ± 13
**7d**	4-CH_3_	126 ± 5
**7e**	4-OCF_3_	158 ± 6
**7f**	4-Br	123 ± 2
**7g**	4-Cl	142 ± 9
**7h**	H	107 ± 9
**7i**	4-CF_3_	162 ± 8
**7j**	3-Cl	135 ± 5
**7k**	3-Br	118 ± 5
**7l**	2-Cl	130 ± 3
**7m**	2-F	157 ± 4
**7n**	3-F	156 ± 7
**7o**	2-Br	114 ± 5
**7p**	3-CH3	109 ± 8
**7q**	4-CH(CH_3_)_2_	139 ± 2
**7r**	4-C(CH_3_)_3_	142 ± 3
**7s**	2-naphthyl	133 ± 2
**7t**	2-CF_3_	150 ± 6
**7u**	3-CF_3_	157 ± 4
**7v**	2-CH_3_	115 ± 5
**7w**	3,5-diF	152 ± 7
**7x**	3,5-diCl	144 ± 4
**7y**	2,4-diF	191 ± 6
**7z**	2,6-diCl	125 ± 4
**7aa**	2,4-diCl	145 ± 5
**7ab**	2,5-diF	145 ± 9
**7ac**	3-F-4-CH_3_	123 ± 6
**7ad**	2,5-diCH_3_	125 ± 4

**Table 2 ijms-23-07959-t002:** The effects of derivatives **8** and **8a**–**8ag** on melanogenesis in B16 cells (50 μM).

Derivative	Substituent (R1)	Relative Melanin Content (%)
NC	-	100 ± 9
8-MOP	-	113 ± 10
**8**	-	305 ± 49
**8a**	4-F	127 ± 11
**8b**	4-OCH_3_	103 ± 10
**8c**	4-Ph	112 ± 9
**8d**	4-CH_3_	105 ± 1
**8e**	4-OCF_3_	117 ± 3
**8f**	4-Br	111 ± 5
**8g**	4-Cl	109 ± 4
**8h**	H	109 ± 5
**8i**	4-CF_3_	122 ± 3
**8j**	3-Cl	105 ± 13
**8k**	3-Br	102 ± 8
**8l**	2-Cl	108 ± 10
**8m**	2-F	112 ± 24
**8n**	3-F	114 ± 9
**8o**	2-Br	103 ± 3
**8p**	3-CH_3_	96 ± 9
**8q**	4-NO_2_	85 ± 6
**8r**	4-C(CH_3_)_3_	114 ± 6
**8s**	2-naphthyl	111 ± 9
**8t**	3,5-diCl	115 ± 5
**8u**	3-CF_3_	121 ± 10
**8v**	2-CH_3_	97 ± 1
**8w**	3-NO_2_	93 ± 15
**8x**	3-OCH_3_	95 ± 16
**8y**	3-OCF_3_	123 ± 7
**8z**	3,5-diCF_3_	134 ± 26
**8aa**	2-Cl-6-F	128 ± 13
**8ab**	3,4-diCl	124 ± 13
**8ac**	3,5-diF	127 ± 6
**8ad**	3,4-diF	125 ± 2
**8ae**	3,4,5-triF	138 ± 33
**8af**	cyclohexyl	120 ± 7
**8ag**	2-furyl	194 ± 9

## Data Availability

Not applicable.

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
