# Peer review of "Study of Novel Furocoumarin Derivatives on Anti-Vitiligo Activity, Molecular Docking and Mechanism of Action"

_ijms, 2022, doi:10.3390/ijms23147959_

Round 1

Reviewer 1 Report

The manuscript describes the identification of sixty-five molecules able to induce melanin synthesis in a murine B16 cell model. Two compounds have been further examined for their potential action on melanogenesis-related genes and signaling pathways, and rationalization attempts have been provided, on the basis of molecular docking results. While the study is technically sound, and the conclusion is supported by a solid set of experiments, I would recommend to accept the manuscript only after the following corrections.

1. The manuscript conveys an overall impression of unfinished and careless writing. The placeholder text from the article template has not been removed in the introduction and experimental parts, underlining the absence of a proper proofreading process. A very great number of grammatical and typographical mistakes makes the text hardly readable (e.g. in the abstract, the use of ‘was’ instead of ‘is’ in the first line and multiple spaces missing, etc.; page 21, ‘reactionwascomplete’, ‘The organic layer    was’, ‘ether/ethylacetate’, ‘benzene sulfonate7a-7ad’). A complete rework of the text is needed.

2. In Tables 1 and 2, and in the text, the significant figures should be revised, since no standard error should contain more than one significant digit. Additionally, main values should be rounded accordingly (e.g. “126±9” instead of “125.6±8.113” for 7, cf Table 1).

3. The majority of reported experiments involved B16 cells, which are melanoma cells from mice. The study would be more thorough with an assay investigating the potential of these compounds in normal melanocytes and/or in human systems. The authors should either include such an experiment or include a perspective in their conclusion.

Author Response

Dear reviewer:

    First we would like to thank you for giving us opportunity to revise our manuscript (MS No.: ijms-1781859; Title: "Study of Novel Furocoumarin Derivatives on Anti-Vitiligo Activity, Molecular Docking and Mechanism of Action"). Your professional advice has been of great use to us. And the ms was carefully revised as you suggested in following responses to ensure it follows the standards of the journal. Looking forward to the next cooperation!

                                     Sincerely Yours    Haji Akber Aisa    2022.6.28

The manuscript describes the identification of sixty-five molecules able to induce melanin synthesis in a murine B16 cell model. Two compounds have been further examined for their potential action on melanogenesis-related genes and signaling pathways, and rationalization attempts have been provided, on the basis of molecular docking results. While the study is technically sound, and the conclusion is supported by a solid set of experiments, I would recommend to accept the manuscript only after the following corrections.

  1. The manuscript conveys an overall impression of unfinished and careless writing. The placeholder text from the article template has not been removed in the introduction and experimental parts, underlining the absence of a proper proofreading process. A very great number of grammatical and typographical mistakes makes the text hardly readable (e.g. in the abstract, the use of ‘was’ instead of ‘is’ in the first line and multiple spaces missing, etc.; page 21, ‘reactionwascomplete’, ‘The organic layer   was’, ‘ether/ethylacetate’, ‘benzene sulfonate7a-7ad’). A complete rework of the text is needed.

Response: The language and style of ms was checked and revised carefully. Extra placeholder text were removed and missing space were added as well.

  1. In Tables 1 and 2, and in the text, the significant figures should be revised, since no standard error should contain more than one significant digit. Additionally, main values should be rounded accordingly (e.g. “126±9” instead of “125.6±8.113” for 7, cf Table 1).

Response: The main values and significant figures in Table 1 and 2 were all revised as you suggested.

  1. The majority of reported experiments involved B16 cells, which are melanoma cells from mice. The study would be more thorough with an assay investigating the potential of these compounds in normal melanocytes and/or in human systems. The authors should either include such an experiment or include a perspective in their conclusion.

Response: The melanogenesis effect of two compounds on human normal melanocytes (PIG3V) were added in Figure 2 and 3 as suggested.

Reviewer 2 Report

The reviewed article presents the studies of novel furocoumarin derivatives in terms of their effect on melanogenesis. The information presented has great scientific potential and would certainly arouse interest among the journal readers. However, in my opinion, the work requires some changes and improvements before publishing.

1. There are editing and linguistic errors at work – additional revision is required.

2. I don't know why the individual sections contain guidelines for the authors.

3. Moreover, the introduction to the publication is too long and partly too detailed. It contains several figures from other publications. I do not believe that it is necessary to copy the figures in the introduction to the research article.

4. The work includes figures and shames. I suggest standardizing this nomenclature.

5. With such small and low-quality microscopic images, the reader is not able to notice any morphology of the cells.

6. Fig. 10 - How to explain the increase in melanin while the effect on TYR activity is not observed?

7. Please standardize the presentation of the statistical analysis in the figures (e.g. 10 and 11)

8. Docking studies – please provide binding affinity data.

9. The analysis of CXCL10 secretion – The analogous experiments should be performed for cells treated only with 7y or 8 compounds.

10. Fig. 17 and 18 - Were the obtained results calculated to b-actin level? If not, this issue should be corrected. If so, please change the description of the y-axis in the bar graphs.

11. I strongly suggest separating Results and Discussion.

12. It is a shame that the Authors did not decide to test chosen two compounds using human normal melanocytes. I am afraid the regulation of melanogenesis in melanoma cells does not represent cytophysiological conditions.

13. With so much information in the methodology (mainly the chemical part), it is worth considering providing detailed NMR data in supplementary materials.

14. In my opinion, the presentation of the results for 7y and 8 compounds in different figures makes it difficult to compare them.

Author Response

Dear reviewer:

First we would like to thank you for giving us opportunity to revise our manuscript (MS No.: ijms-1781859; Title: "Study of Novel Furocoumarin Derivatives on Anti-Vitiligo Activity, Molecular Docking and Mechanism of Action"). Your professional advice has been of great use to us. And the ms was carefully revised as you suggested in following responses to ensure it follows the standards of the journal. Looking forward to the next cooperation!

                                     Sincerely Yours Haji Akber Aisa  2022.6.28

The reviewed article presents the studies of novel furocoumarin derivatives in terms of their effect on melanogenesis. The information presented has great scientific potential and would certainly arouse interest among the journal readers. However, in my opinion, the work requires some changes and improvements before publishing.

  1. There are editing and linguistic errors at work – additional revision is required.

Response: The language and style of ms were checked and revised carefully as you suggested.

  1. I don't know why the individual sections contain guidelines for the authors.

Response: It is sorry for the negligence, the guideline in introduction part was removed.

  1. Moreover, the introduction to the publication is too long and partly too detailed. It contains several figures from other publications. I do not believe that it is necessary to copy the figures in the introduction to the research article.

Response: The “Introduction section” was simplified and all these figures were deleted, the corresponding references and other figures were modified as well.

  1. The work includes figures and shames. I suggest standardizing this nomenclature. (In my opinion, all graphics can be identified as figures. Introducing the term "scheme" makes it a bit difficult to follow in the manuscript.)

Response: The Scheme 1 was standardized as Figure 1, and other Figures were modified as well.

  1. With such small and low-quality microscopic images, the reader is not able to notice any morphology of the cells.

Response: The original images were substituted by High quality ones as shown in Figure 2 and 3.

  1. Fig. 10 - How to explain the increase in melanin while the effect on TYR activity is not observed?

Response: Actually, compound 7y had a lesser stimulation effect (114.52% for 1 1 μM; 114.61% for 10 μM) on TYR activity, since when compared with control (100.0%).  

  1. Please standardize the presentation of the statistical analysis in the figures (e.g. 10 and 11) (Ad. 2 e.g. fig. 11 DE presents the connecting proper bars between treated samples and the untreated control. In turn, only signs of statistical significance (*) above the error bars are observed in Fig.10 DE. Taking into account the fact that the performed statistical analysis compares the individual tests with the control, the connecting bars may be removed)

Response: The connecting bars was deleted in Figure 3 (Original Figure 11

  1. Docking studies – please provide binding affinity data.

Response: The Score, Free Binding Energies (ΔG) and root-mean-square deviations (RMSD) for the best pose of compound 7y and 8 within JAK1 and JAK2 were listed in the following table.

Targets

Compounds

Score

RMSD

(Å)

Binding energy_ΔG

(kcal/mol)

-CDOCKER_

ENERGY

(kcal/mol)

-CDOCKER_

INTERACTION

_ENERGY

(kcal/mol)

JAK1

7y

25.3086

48.9295

1.16

-26.4245

JAK2

7y

28.5653

49.8894

1.03

-30.5082

JAK1

8

29.0854

46.4974

1.16

-26.4245

JAK2

8

26.2908

39.9148

1.03

-25.2251

  1. The analysis of CXCL10 secretion – The analogous experiments should be performed for cells treated only with 7y or 8 compounds.

Response: The analysis of CXCL10 secretion for cells treated only with 7y or 8 compounds by ELISA kit and the result were listed in the following table.

Compounds

CXCL10 (pg/ml)

NC

3.012±0.175

7y

2.939±0.749

8

3.025±0.218

No obvious effect of compounds on CXCL10 secretion were observed, probably because the JAK-STAT signaling was not activated by IFN-γ.

  1. Fig. 17 and 18 - Were the obtained results calculated to b-actin level? If not, this issue should be corrected. If so, please change the description of the y-axis in the bar graphs.

Response: The y-axis in the bar graphs were changed as Reviewer suggested.

  1. I strongly suggest separating Results and Discussion.

Response: The“Results”and “Discussion” sections were separated.

  1. It is a shame that the Authors did not decide to test chosen two compounds using human normal melanocytes. I am afraid the regulation of melanogenesis in melanoma cells does not represent cytophysiological conditions.

Response: The melanogenesis effect of two compounds on human normal melanocytes (PIG3V) were added in Figure 2 and 3.

  1. With so much information in the methodology (mainly the chemical part), it is worth considering providing detailed NMR data in supplementary materials.

Response: The NMR date and other chemical characterizations of 7a-7ad, 8a-8ag were moved to Supplementary Materials file.

  1. In my opinion, the presentation of the results for 7y and 8 compounds in different figures makes it difficult to compare them.

Response: It was so shame that these two serious of compounds were synthesized with two month gap in between. However, the concentration-dependent effect of compound 8 seems to be more pronounced.

Reviewer 3 Report

This is a very interesting paper, I have so far thought that furocoumarins/psoralens act on melanogenesis mainly as non-specific stimulators via photysensitizing the cells. For me interactions with signalling pathways is a relative novelty (perhaps not unexpected), and I strongly suggest the authors following this way. I have, anyway, some quite serious remarks which must be addressed on the revision:

1. Please very carefully check the text from the point of view of technical errors. All the chapters contain an "Introduction" with fragments of the instruction for authors, and a number of words are joined ("glued"). This may be a result of computer system incompatibility but at the same time it is a symptoms of carelessness when preparing the manuscript. The text also contains numerous spelling/grammar errors which must be corrected. I rarely start with technical errors but these are really bad.

2. Please check spacing. In particular, the abbreviation of the unit should be separated from the number if the first starts with a letter (e.g. 50 nm). If there is no letter after the digital, there should be no space (e.g. 10x, 40 C).

3. There is a wrong view on the engagement of tyrosinase in the Raper-Mason pathway of melanogenesis. DOPA is NOT an intermediate between Tyrosine and dopaquinone. I.e. it is not a product of tyrosinase. DOPA appears later on in the redox exchange reaction and then it may activate met-tyrosinase and serve as a substrate, as well; for details see: Plonka et al. BJD 2006  10.1111/j.1365-2133.2006.07376.x; Land et al., Method Enzymol, 2004, doi: 10.1016/S0076-6879(04)78005-2; Schallreuter et al. EXD 2008, 10.1111/j.1600-0625.2007.00675.x

4. For the same reason the authors should precise which enzymatic activity of tyrosinase  they measured (actually it was the catecholase activity). Please discuss the enzymatic activity from this point of view. For details see: Winder & Harris, Eur J Biochem 1991, 10.1111/j.1432-1033.1991.tb16018.x

5. The B16 cells were kept in DMEM, which itself may affect (stimulate) melanogenesis. Please discuss. See e.g. Slominski et al., PCMR 2012, 10.1111/j.1755-148X.2011.00898.x

6, 7. Please discuss the results in the context of eu/pheomelanogenesis regulation (perhaps you have some data?), and also in the context of phototoxic actions of furocoumarins mentioned in the beginning.

Author Response

Dear reviewer:

First we would like to thank you for giving us opportunity to revise our manuscript (MS No.: ijms-1781859; Title: "Study of Novel Furocoumarin Derivatives on Anti-Vitiligo Activity, Molecular Docking and Mechanism of Action"). Your professional advice has been of great use to us. And the ms was carefully revised as you suggested in following responses to ensure it follows the standards of the journal. Looking forward to the next cooperation!

                                     Sincerely Yours  Haji Akber Aisa  2022.6.28

This is a very interesting paper, I have so far thought that furocoumarins/psoralens act on melanogenesis mainly as non-specific stimulators via photysensitizing the cells. For me interactions with signalling pathways is a relative novelty (perhaps not unexpected), and I strongly suggest the authors following this way. I have, anyway, some quite serious remarks which must be addressed on the revision:

  1. Please very carefully check the text from the point of view of technical errors. All the chapters contain an "Introduction" with fragments of the instruction for authors, and a number of words are joined ("glued"). This may be a result of computer system incompatibility but at the same time it is a symptoms of carelessness when preparing the manuscript. The text also contains numerous spelling/grammar errors which must be corrected. I rarely start with technical errors but these are really bad.

Response: It was so sorry for our carelessness. All the instructions for author were removed. Extra spaces were deleted and the ms was checked, revised again very carefully.

  1. Please check spacing. In particular, the abbreviation of the unit should be separated from the number if the first starts with a letter (e.g. 50 nm). If there is no letter after the digital, there should be no space (e.g. 10x, 40C).

Response: In Manuscript and Supplementary Materials section, All the improper spacing between number and unit, were modified.

  1. There is a wrong view on the engagement of tyrosinase in the Raper-Mason pathway of melanogenesis. DOPA is NOT an intermediate between Tyrosine and dopaquinone. I.e. it is not a product of tyrosinase. DOPA appears later on in the redox exchange reaction and then it may activate met-tyrosinase and serve as a substrate, as well; for details see: Plonka et al. BJD 2006  10.1111/j.1365-2133.2006.07376.x; Land et al., Method Enzymol, 2004, doi: 10.1016/S0076-6879(04)78005-2; Schallreuter et al. EXD 2008, 10.1111/j.1600-0625.2007.00675.x

Response: The incorrect description for the L-Dopa was revised and the corresponding references ([10] and [11]) you suggested were cited as well. 

  1. For the same reason the authors should precise which enzymatic activity of tyrosinase they measured (actually it was the catecholase activity). Please discuss the enzymatic activity from this point of view. For details see: Winder & Harris, Eur J Biochem 1991, 10.1111/j.1432-1033.1991.tb16018.x

Response: The L-dopa and tyrosine are both able to bind to the separate active site of tyrosinase as substrate, and oxidized to dopaquinone. In our research, the catechols activity of tyrosinase (L-dopa as substrate) were studied. And it was described in this point of view.

  1. The B16 cells were kept in DMEM, which itself may affect (stimulate) melanogenesis. Please discuss. See e.g. Slominski et al., PCMR 2012, 10.1111/j.1755-148X.2011.00898.x

Response: As the reference described, the L-tyrosine could be formed from L- L-phenylalanine catalyzed by PAH (phenylalanine hydroxylase). Considering the  low content of L-phenylalanine in DMEM, the blank control group was set in our experiments (described as NC; And the ODs of other groups were all normalized with the NC group in order to avoid the errors). However, could you give us some advice for a more suitable medium to improve our activity evaluation, since some other medium were Unsatisfactory in our previous experiments.

6, 7. Please discuss the results in the context of eu/pheomelanogenesis regulation (perhaps you have some data?), and also in the context of phototoxic actions of furocoumarins mentioned in the beginning.

Response: The melanin content mentioned in our research represented the total pigment, including eumelanins and pheomelanins. Since the precise synthetic mechanisms contributing to the formation of these pigments were really complicated and not known, no more further experiments were performed. The melanin content was the activity index for primary screening in our lab. Besides, we would like to learn related theory and method to perfecting our studies.

For the phototoxic actions of furocoumarins, it is with great regret there was no suitable experiments for it in our team, and related content in ms was deleted as well. The toxicity of furocoumarins to cells was evaluated by CCK-8 method at present. We would appreciate it if you could give some advice on how to evaluate phototoxic actions of furocoumarins. 

Round 2

Reviewer 1 Report

The requested corrections have been made, and the addition of experiments performed on a human system (the vitiligo model PIG3V) adds some value to the study. Therefore I would recommend to accept the manuscript for publication.

Reviewer 3 Report

All the suggestions have been considered and the paper may be published in this form. As to the questions: 5. - The control media suggested as proper for controls are RPMI and Ham's F10  buffore according to the cited Slominski et al.) As to 7. - there may be planed additional experiments with the detection of singlet oxygene and lipid peroxidation under light in the presence and absence of furocoumarins in the medium.